# Human Umbilical Cord Mesenchymal Stem Cell-Derived Exosomes Rescue Testicular Aging

**DOI:** 10.3390/biomedicines12010098

**Published:** 2024-01-03

**Authors:** Peng Luo, Xuren Chen, Feng Gao, Andy Peng Xiang, Chunhua Deng, Kai Xia, Yong Gao

**Affiliations:** 1Department of Urology and Andrology, The First Affiliated Hospital, Sun Yat-sen University, Guangzhou 510080, China; luop39@mail.sysu.edu.cn (P.L.); chenxr73@mail2.sysu.edu.cn (X.C.); gaof65@mail2.sysu.edu.cn (F.G.); dengchh@mail.sysu.edu.cn (C.D.); 2Center for Stem Cell Biology and Tissue Engineering, Key Laboratory for Stem Cells and Tissue Engineering, Ministry of Education, Sun Yat-sen University, Guangzhou 510080, China; xiangp@mail.sysu.edu.cn; 3Reproductive Medicine Center, The Key Laboratory for Reproductive Medicine of Guangdong Province, The First Affiliated Hospital, Sun Yat-sen University, Guangzhou 510080, China; 4Maoming Maternal and Child Health Hospital, Maoming 525000, China; 5Department of Biochemistry, Zhongshan School of Medicine, Sun Yat-sen University, Guangzhou 510080, China

**Keywords:** testicular aging, mesenchymal stem cell, exosome, testosterone, spermatogenesis, macrophage

## Abstract

Background: Testicular aging is associated with diminished fertility and certain age-related ailments, and effective therapeutic interventions remain elusive. Here, we probed the therapeutic efficacy of exosomes derived from human umbilical cord mesenchymal stem cells (hUMSC-Exos) in counteracting testicular aging. Methods: We employed a model of 22-month-old mice and administered intratesticular injections of hUMSC-Exos. Comprehensive analyses encompassing immunohistological, transcriptomic, and physiological assessments were conducted to evaluate the effects on testicular aging. Concurrently, we monitored alterations in macrophage polarization and the oxidative stress landscape within the testes. Finally, we performed bioinformatic analysis for miRNAs in hUMSC-Exos. Results: Our data reveal that hUMSC-Exos administration leads to a marked reduction in aging-associated markers and cellular apoptosis while promoting cellular proliferation in aged testis. Importantly, hUMSC-Exos facilitated the restoration of spermatogenesis and elevated testosterone synthesis in aged mice. Furthermore, hUMSC-Exos could attenuate inflammation by driving the phenotypic shift of macrophages from M1 to M2 and suppress oxidative stress by reduced ROS production. Mechanistically, these efficacies against testicular aging may be mediated by hUMSC-Exos miRNAs. Conclusions: Our findings suggest that hUMSC-Exos therapy presents a viable strategy to ameliorate testicular aging, underscoring its potential therapeutic significance in managing testicular aging.

## 1. Introduction

The global demographic shift toward an aged population precipitates the increased prevalence of age-related diseases, demanding heightened research attention [1]. The gonad is particularly susceptible to aging-associated detriments, resulting in marked fertility and endocrine alterations that perturb multisystemic homeostasis [2]. While ovarian aging has been extensively studied, our understanding of male testicular aging is comparatively nascent [3]. Beyond fertility decline and sexual dysfunction, testicular aging manifests systemically as sleep disturbances, osteoporosis, and centripetal obesity [4,5,6]. Moreover, this aging process exhibits associations with diabetes and cardiovascular pathologies [7,8]. However, there are currently limited treatment options for testicular aging and its manifestations, and the commonly used treatment is exogenous testosterone replacement therapy (TRT). TRT is a prevalent intervention for testosterone deficiencies secondary to testicular aging, but TRT disrupts the hypothalamic–pituitary–gonadal (HPG) axis and further hinders spermatogenesis. Moreover, prolonged TRT has been correlated with a series of adverse effects, encompassing infertility, polycythemia, lipid metabolism anomalies, stroke, and even prostate cancer [9,10]. This backdrop emphasizes the pressing need for alternative therapeutic avenues for testicular aging.

In recent years, stem cell therapy has emerged as a potent method to combat age-related diseases, circumventing some of the drawbacks of TRT [11,12]. While promising, stem cell therapy is often associated with the risks such as microvasculature occlusion, potential malignant transformations, immune rejection, or the establishment of enduring grafts post-administration [13,14,15]. Moreover, a multitude of ethical quandaries associated with stem cell therapy still need breakthroughs [16]. Given these dilemmas, a promising, and potentially more cost-effective, alternative therapy to traditional cell-based interventions is necessary. In the realm of regenerative medicine for aging, exosomes have emerged as a focal point of interest [17]. Exosomes are the most extensively researched subtype of extracellular vesicles and are laden with active components, such as lipids, proteins, nucleic acids, and cell-derived metabolites [18]. Of particular significance are the exosomes emanating from mesenchymal stem cells (MSC-Exos). Their impressive secretory prowess and their capacity to orchestrate tissue regeneration, modulate immune responses, and facilitate cellular repair render them a topic of intense investigation [19]. At present, MSC-Exos are recognized as the predominant adult stem cell-derived extracellular vesicles employed in anti-aging studies [20,21]. In osteoporosis, the efficacy of MSC-Exos is significant, improving osteoblast function and bone morphology [22]. Moreover, their therapeutic versatility extends to neuroprotection, as evidenced by their ability to protect dopaminergic neurons, thereby enhancing dopamine concentration in the striatum and offering potential interventions for Parkinson’s disease [23]. In Alzheimer’s disease, MSC-Exos therapy is also becoming a promising therapeutic approach [24]. Additionally, in the context of premature ovarian failure, MSC-Exos have demonstrated efficacy in revitalizing the ovarian vascular system, promoting follicular growth, and reinstating ovarian functionalities [25]. Compared to stem cells, MSC-Exos are more stable, have no risk of aneuploidy, have a lower chance of immune rejection, and can provide an alternative therapy for various diseases [26]. Given their repertoire, it is imperative to rigorously assess the efficacy of MSC-Exos in relation to testicular aging.

Human umbilical cord mesenchymal stem cells (hUMSCs), which are derived from the Wharton’s jelly of the umbilical cord, possess distinct attributes compared to their counterparts from other sources [27]. Their superior proliferative capacity, coupled with reduced immunogenicity, delineates them as prime candidates for therapeutic interventions [11,28]. Additionally, their sourcing minimizes the infection transmission risks seen with adult-derived cells. The prospect of long-term cryopreservation means that these cells can be stockpiled for future therapeutic applications, further accentuating their prominence in regenerative medicine [29,30]. Exosomes secreted by hUMSCs (hUMSC-Exos) are enriched with a myriad of growth factors, mRNAs, miRNAs, and proteins [31,32]. This molecular milieu positions hUMSC-Exos as potential regulators of inflammation, facilitators of tissue regeneration, and modulators in disease trajectory. Currently, hUMSC-Exos have been widely used in the field of regeneration. Sun et al. showed the promotion of function recovery in mice with spinal cord injury through the intravenous administration of hUMSC-Exos [33]. In addition, Han et al. employed hUMSC-Exos as a therapeutic strategy to alleviate tendon injury [34]. Moreover, Yang et al. reported that hUMSC-derived exosomes alleviated age-related infertility in women [35]. Yet, the exact implications of hUMSC-Exos in the context of testicular aging remain to be delineated.

This study endeavors to elucidate the therapeutic implications of hUMSC-Exos on testicular aging in naturally aged male mice. We aim to discern their potential in ameliorating testicular aging and restoring testicular function, and further reveal the underlying mechanistic pathways. Finally, our study provides a practicable way to rejuvenate aged testis.

## 2. Materials and Methods

### 2.1. Animals

Eighteen-month-old C57BL/6 male mice were purchased from the Model Animal Research Center of GemPharmatech Co., Ltd. (Nanjing, China). All mice were maintained in the animal facility of Sun Yat-sen University under a 12 h light–dark cycle and provided with water and food. The mice were used for the experiments at 22 months of age. All animal experiments were performed according to experimental protocols approved by the ICE for Clinical Research and Animal Trials of the First Affiliated Hospital of Sun Yat-sen University (Approval number: #2022-392; 15 August 2022) and complied with the National Institute of Health Guide for the Care and Use of Laboratory Animals.

### 2.2. Study Design and Experimental Procedure

Twelve aged mice were randomly divided into two groups (*n* = 6 mice/group): the control (CON) group, which received an injection of phosphate-buffered saline (PBS), and the experimental (EXO) group, which received an injection of hUMSC-Exos. The method of intratesticular injection involved the following procedures [36]: Initially, mice were anesthetized via intraperitoneal injection of Avertin (250 mg/kg) (Sigma-Aldrich, St. Louis, MO, USA). Then, the surgical site was disinfected with 75% alcohol and povidone-iodine. Under sterile conditions, we made an incision approximately 2 cm anterior to the genitalia on the ventral skin and body wall using a sterile surgical blade. Subsequently, the testes were exteriorized by gently grasping the fat pad, with care taken to handle the tissues delicately to avoid damaging blood vessels. Each testis was secured with fine forceps, and a 33-gauge needle (Hamilton, Bonaduz, Switzerland) was used to inject hUMSC-Exos (20 μL/testis; number: 10^10^ particles/testis; concentration: 5.0 × 10^11^ particles/mL) or PBS (20 μL/testis) along the longitudinal axis of the testis. Thereafter, the incision was closed using absorbable surgical sutures. At 20 d after the first injection, mice in the EXO and CON groups received a second injection similar to the first injection. Finally, the mice were euthanized 20 d after the second intratesticular injection, and the testes, semen, and blood were collected and preserved for subsequent experiments.

### 2.3. Culturing of hUMSCs

hUMSCs were purchased from the National Collection of Authenticated Cell Cultures in China. The hUMSCs were maintained in Dulbecco’s modified Eagle’s medium/F12 (DMEM/F12; Gibco, Grand Island, NY, USA) mixed with 10% fetal bovine serum (FBS; Gibco) in a 5% CO_2_ cell incubator at 37 °C. The study and all experiments involving cells derived from human umbilical cord tissue conformed to the Declaration of Helsinki.

### 2.4. Preparation of Exosomes

As previous report [37], hUMSCs at passage 3 were seeded in T-75 cm^2^ flask and grown to approximately 80% confluence with 3 × 10^6^ cells. After discarding the cell medium and rinsing the cells three times with PBS, hUMSCs were incubated in DMEM/F12 containing 10% exosome-depleted FBS (System Biosciences, Palo Alto, CA, USA) for 48 h at 37 °C in a 5% CO_2_ atmosphere and cell supernatant was further collected for exosome extraction.

Method for exosome isolation was modified from the previous report [38]. Briefly, dead cells, cellular debris, and large extracellular vesicles, excluding exosomes, were discarded by centrifugation at 300× *g* for 10 min, followed by centrifugation at 10,000× *g* for 30 min. Subsequently, the supernatant was ultracentrifuged at 100,000× *g* to obtain the exosomes. After filtration through a 0.22 μm syringe filter (Millipore, Darmstadt, Germany), the supernatants were ultracentrifuged at 100,000× *g* for 1 h on an SW32Ti rotor (Beckman Coulter, Krefeld, Germany) for the initial isolation of exosomes. The exosomes were then immersed in PBS and ultracentrifuged at 100,000× *g* for 1 h. All centrifugation and ultracentrifugation procedures were performed at 4 °C. After serial centrifugation and ultracentrifugation procedures, obtained exosomes were dissolved in 200 μL PBS. The total protein content of exosomes released by 3 × 10^6^ hUMSCs in 48 h was approximately 82.30 ± 4.06 μg, quantified using a BCA Protein Assay, and the particle concentration was 11.97 ± 0.47 × 10^11^ particles/mL, quantified using nanoparticle tracking analysis (NTA; Nanosight NS300, Malvern Instruments, Worcestershire, UK). The exosomes were diluted in PBS and frozen at −80 °C for further use.

### 2.5. Identification and Characterization of Exosomes

#### 2.5.1. Transmission Electron Microscopy

The morphology of the hUMSC-Exos was visualized using transmission electron microscopy (TEM; FEI Tecnai 12, Eindhoven, The Netherlands). Briefly, 20 μL of hUMSC-Exos was placed on a carbon gel, incubated for 5 min, and then subjected to negative staining with 2% uranyl acetate, air-dried, and imaged.

#### 2.5.2. Nanoparticle Tracking Analysis

hUMSC-Exos stored at −80 °C were resuspended in PBS, and the size distribution and concentration of exosomes were measured using NTA. Referring to several examples in the literature [39,40,41], we set the camera level to 14 and the detection threshold to 3 in this study. The chosen camera level ensured visibility of all particles, and the threshold was set at a reasonable value to minimize non-particle background noise without excluding certain particles. In this study, the peak particle size of hUMSC-Exos was approximately 139.6 nm. Finally, hUMSC-Exos were diluted to 5 × 10^11^ particles/mL prior to use.

#### 2.5.3. Western Blotting

The expression of the exosomal markers CD9 and TSG101 was assessed using Western blot analysis, as described previously [18,38]. Briefly, proteins were separated on a 12% SDS-PAGE gel (Invitrogen, Carlsbad, CA, USA) and transferred to a polyvinylidene fluoride membrane (Invitrogen) for immunoblotting with specific primary antibodies. The primary antibodies used were anti-CD9 (Abcam, Boston, MA, USA), anti-TSG101 (ABclonal, Wuhan, China), and anti-calnexin (ABclonal, Wuhan, China), with CALNEXIN as the control. Appendix A lists all the antibody information. The blots were imaged using ImageJ V1.8.0 (National Institutes of Health, Bethesda, MD, USA).

### 2.6. Exosomal Labeling In Vitro

Before injection of HUMSC-Exos into the testes of aged mice, the HUMSC-Exos were stained with the molecular probe CM-Dil Dye (Invitrogen), according to the manufacturer’s protocol. Briefly, hUMSC-Exos were incubated with 1 μL of CM-Dil dye (1 mM) diluted in 1 mL of PBS for 20 min at 37 °C. The stained exosomes were washed with PBS, and the free dye was removed by ultrafiltration.

### 2.7. SA-β-gal Assay

SA-β-gal (senescence-associated beta-galactosidase) staining of frozen testicular tissues was performed using a Senescence β-Galactosidase Staining Kit (Beyotime Biotechnology, Shanghai, China), according to the manufacturer’s protocol. Nuclear Fast Red staining (Beyotime Biotechnology) was performed after SA-β-gal staining, and the tissues were photographed using a Leica DMi8 microscope (Leica, Wetzlar, Germany). ImageJ V1.8.0 (National Institutes of Health) was used to segment the pictures based on color difference, extracting the positively stained area of SA-β-gal, and quantifying the occupied area in each vision.

### 2.8. Testosterone Concentration Measurement

Serum samples were collected from aged mice at designated time points and stored at −80 °C for testosterone concentration measurement using a chemiluminescent immunoassay (CLIA) system (KingMed Diagnostics Group Co., Ltd., Guangzhou, China). Briefly, 100 μL of each sample was aliquoted for testosterone level measurements, and the kit was then inserted into a CLIA system for analysis. The lower limit of detection for the CLIA system was 0.01 ng/mL. The within-assay CV for the CLIA system ranged from 1.8 to 5.0%, and the inter-assay CV ranged from 2.4 to 5.1%, according to the manufacturer’s information.

### 2.9. Hematoxylin and Eosin (H&E) Staining

Mice testes were collected and fixed in Bouin’s solution (Sigma-Aldrich) overnight, and the tibialis anterior muscle was collected and fixed in PFA (Sigma-Aldrich) overnight. Thereafter, the testes and tibialis anterior were dehydrated in 75% ethanol, embedded in paraffin, and cut into 4 μm sections. Prior to staining, all the tissue sections were deparaffinized in xylene (Sigma-Aldrich), hydrated in decreasing concentrations of ethanol, rinsed in PBS, and stained with hematoxylin and eosin (H&E; Sigma-Aldrich) for histological analysis. After staining, the sections were photographed using a Leica DMi8 microscope (Leica). As mentioned in previous reports [42,43], the thickness of the spermatogenic epithelium was measured from the outer edge of the tubule to the nucleus of the elongating spermatids in the inner edge of the tubule using ImageJ V1.8.0 software (National Institutes of Health). For one seminiferous tubule, the thicknesses of three different positions were randomly recorded and their average value was calculated as the final thickness.

### 2.10. Grip Strength Measurement

At 20 d after the second injection of hUMSC-Exos into the testis, the maximum value of forelimb grip strength of the aged mice was measured using a Columbus Instruments Grip Strength Meter (Columbus Instruments, Columbus, OH, USA), according to the manufacturer’s instructions. Five measurements taken at least 30 min apart were performed for each mouse, and maximum values of forelimb grip strength in each measurement were recorded.

### 2.11. Immunofluorescence Staining

Mouse testicular tissues were fixed with 4% PFA (Sigma-Aldrich) at room temperature (RT; 20–25 °C) for 2 h, followed by dehydration using a sucrose gradient (10, 20, and 30%). Subsequently, the tissues were frozen in a cryostat (Leica CM1950, Wetzlar, Germany) and cut into 10 μm thick sections. After washing three times with PBS, the sections were permeabilized with 0.5% Triton X-100 (Sigma-Aldrich) for 20 min, blocked with 3% BSA (Sigma-Aldrich) at RT for 45 min, and incubated with the primary antibody overnight at 4 °C. Thereafter, the sections were incubated in the dark with suitable secondary antibodies at RT for 1 h and stained with 4,6-diamidino-2-phenylindole (DAPI; Invitrogen, Carlsbad, CA, USA) for 5 min. After washing three times with PBS, the tissues and fluorescent signals were imaged using an LSM800 confocal microscope (Zeiss, Jena, Germany). ImageJ V1.8.0 software (National Institutes of Health) was used to quantify the fluorescence signal. In the images, fluorescence signal of positive cells should be bright and uniform, while the fluorescence signal of negative cells should be absent or weak. DAPI was used to indicate the nucleus. Appendix A lists the primary and secondary antibodies.

### 2.12. Computer-Aided Sperm Analysis (CASA)

CASA was performed as previously reported [44]. Briefly, the cauda epididymis on both sides was removed from the mice and cut open with small scissors. To induce sperm release, each cauda epididymis was incubated in 500 μL of DMEM/F12 containing 0.5% BSA (Sigma-Aldrich) at 37 °C for 20 min. Afterwards, the cauda epididymis was discarded, and the sperm was analyzed using Hamilton Thorne’s Ceros II system (Hamilton Thorne, Beverly, MA, USA). At least six fields were evaluated for each semen sample, and the sperm concentration and proportions of motile and progressively motile spermatozoa were determined.

### 2.13. Analysis of Macrophage Phenotype Using Flow Cytometry

Testicular tissues were enzymatically dissociated and analyzed, as previously reported [45]. Briefly, tunica albuginea was carefully removed from the testis and discarded, and cells in the testicular interstitium were lysed using 1 mg/mL of type IV collagenase (Gibco, Grand Island, NY, USA) in DMEM/F12 (Gibco) for 20 min with shaking at a slow speed (90 cycles/min) at 37 °C. The enzymatic reaction was terminated by the addition of DMEM/F12 (Gibco) containing 10% FBS. (Gibco). Subsequently, the testicular specimens were filtered with a 45 μm syringe filter and centrifuged at 1200 rpm for 4 min to eliminate seminiferous tubules. The samples were immersed in PBS containing 0.1% BSA (Sigma–Aldrich) to prepare a single-cell suspension. Finally, the cells were incubated with antibodies against CD206-APC (Biolegend, San Diego, CA, USA) and CD86-PeCy7 (Biolegend) for 30 min at 37 °C, and the fluorescence intensity of the antibody signal was determined using fluorescence-activated cell sorting (FACS; Becton Dickinson, San Diego, CA, USA). Appendix A lists all the antibodies used for flow cytometry.

### 2.14. Quantitative Real-Time Polymerase Chain Reaction (qRT-PCR)

Total RNA was extracted from mouse tissues using the RNeasy kit (Qiagen, Germantown, MD, USA), according to the manufacturer’s instructions. RNA concentration was assayed using a NanoDrop 1000 spectrophotometer (Thermo Fisher, Waltham, MA, USA), and reverse transcription was performed using the NovoScript^®^ First Strand cDNA Synthesis Kit (Novoprotein, Shanghai, China). The qRT-PCR analysis was performed on a Light Cycler 480 Detection System (Roche, Indianapolis, IN, USA) using the Thunderbird SYBR qPCR Mix (Toyobo, Osaka, Japan). Additionally, a melting curve was generated to verify the presence of individual peaks and to exclude the presence of nonspecific products or primer dimers. The expression levels of the target genes were calculated using the Ct method and normalized to that of β-actin. The primer sequences and the CT values of all genes in the present study are listed in Appendix A.

### 2.15. Reactive Oxygen Species (ROS) Analysis

ROS levels were measured using a Reactive Oxygen Species assay kit (Thermo Fisher Scientific), according to the manufacturer’s instructions. Briefly, cells were collected from the testicular interstitium after enzymatic digestion and incubated in the dark with DMEM/F12 containing 1 μL of dihydroethidium (DHE) for 30 min at 37 °C. After rinsing with PBS, the fluorescence intensity (excitation = 495 nm, emission = 520 nm) of the cells was determined using FACS (Becton Dickinson and Company, Franklin Lakes, NJ, USA).

### 2.16. Bioinformatic Analysis

The hUMSC-Exos miRNA expression microarray GSE69909 was downloaded from the GEO database. Firstly, the significantly enriched miRNAs in hUMSC-Exos were presented using volcano plot and heatmap. Subsequently, TarBase and ElMMo were employed to predict the target genes of the significantly enriched miRNAs in hUMSC-Exos. All predicted targets with prediction scores ≥ 80 underwent subsequent Gene Ontology (GO) analysis and Kyoto Encyclopedia of Genes and Genomes Pathway (KEGG) analysis to investigate the underlying mechanism. Detailed data are listed in Appendix A.

### 2.17. Statistical Analysis

All data were analyzed using GraphPad Prism v8 (GraphPad Software, La Jolla, CA, USA). Comparisons between groups were performed using Student’s *t*-test, one/two-way analysis of variance (ANOVA), or nonparametric tests (Mann–Whitney U tests), and statistical significance was set at *p* < 0.05 (* *p* < 0.05, ** *p* < 0.01, *** *p* < 0.001, and ns = no significance). All data are expressed as the mean ± standard error of the mean (sem) of at least three independent replicate experiments.

## 3. Results

### 3.1. Isolation and Characterization of hUMSC-Exos

The technological process of exosome isolation is shown in Figure 1A. After isolation, hUMSC-Exos were, respectively, characterized using TEM, NTA, and Western blotting. HUMSC-Exos displayed the typical characteristics of exosomes, including a cup-like morphology (Figure 1B), a peak particle size of approximately 139.6 nm (Figure 1C), and the expression of the typical exosomal markers TSG101 and CD9. Subsequently, 20 μL of hUMSC-Exos (number: 10^10^ particles/testis; concentration: 5.0 × 10^11^ particles/mL) was labeled with CM-Dil dye and injected into the testes of aged mice. Most CM-Dil-labeled hUMSC-Exos accumulated in the testicular interstitium, with a small number observed in the outer periphery of the seminiferous ducts 3 days after injection (Figure 1D).

### 3.2. hUMSC-Exos Treatment Ameliorates Cellular Senescence in Aged Testis

After 40 d of hUMSC-Exos treatment, the mice were sacrificed, and the testes were morphologically assessed and sectioned for the analysis of aging-related phenotypes. The hUMSC-Exo-treated aged mice had significantly higher (*p* < 0.001) testicular size and weight index (2.314 ± 0.064% vs. 1.736 ± 0.053%) than those treated with PBS (Figure 2A). However, there was no difference (*p* > 0.05) in body weight between both groups (EXO vs. CON groups: 0.060 ± 1.163 g vs. 0.180 ± 1.157 g) before or after treatment (Appendix A). Immunofluorescence and SA-β-gal staining showed that hUMSC-Exos treatment significantly increased (*p* < 0.001) LAMINB1 expression and decreased (*p* < 0.001) β-galactosidase activity compared with those of the CON group (Figure 2B,C). Overall, these results suggest that hUMSC-Exos treatment alleviates testicular senescence phenotypes.

Terminal deoxynucleotidyl transferase dUTP nick-end labeling (TUNEL) and proliferating cell nuclear antigen (PCNA) staining are typically combined to assess cell regeneration and apoptosis. Compared with that in the CON group, hUMSC-Exos treatment significantly decreased (*p* < 0.001) the number of TUNEL-positive cells in the testes of mice and increased the number of PCNA-positive cells in the seminiferous tubules (Figure 2D,E). These results suggest that hUMSC-Exos treatment can decrease apoptosis and promote cell proliferation in aged testis.

### 3.3. hUMSC-Exos Treatment Improves Spermatogenesis in Aged Mice

Subsequently, we evaluated the effect of hUMSC-Exos treatment on spermatogenesis in aged mice. Similar to previous reports [46], degeneration of the germinal epithelium of seminiferous tubules without apparent spermatogenesis was observed in the CON group (Figure 3A). In contrast, the seminiferous epithelium of mice in the EXO group (59.013 ± 2.539 μm) was significantly thicker (*p* < 0.001) than that of mice in the CON group (33.001 ± 1.229 μm) (Figure 3A). Specifically, multiple layers of spermatogonial stem cells (SSCs) were observed in the seminiferous tubules of mice in the EXO group, with spermatocytes in the basement membrane and many mature spermatozoa on the luminal surface of the seminiferous tubules (Figure 3A). Overall, these data suggest that hUMSC-Exos treatment restores spermatogenesis in aged testis.

As SSCs are crucial for the initiation of spermatogenesis, we further examined the effect of hUMSC-Exos treatment on SSCs. Specifically, immunostaining was performed to examine the expression of stimulation by retinoic acid 8 (STRA8, a marker of undifferentiated SSCs) and DEAD-box helicase 4 (DDX4, a marker of germ cells, including undifferentiated and differentiated SSCs) in testicular tissues. As shown in Figure 3B,C, STRA8- and DDX4-positive cells were significantly higher (*p* < 0.001) in the testes of mice in the EXO group than in those of mice in the CON group, indicating that hUMSC-Exos treatment improved testicular stem cell niche in aged mice and promoted the proliferation and differentiation of recipient SSCs.

Furthermore, peanut agglutinin (PNA) staining was performed to determine whether hUMSC-Exos could promote spermatogenesis in aged mice. To the best of our knowledge, PNA can selectively bind to the acrosome of sperm and can be used as a haploid spermatid marker. Compared with the control group, hUMSC-Exos treatment significantly increased (*p* < 0.001) the number of PNA-positive cells in the seminiferous tubules of the EXO group (Figure 3D), suggesting that hUMSC-Exos treatment can promote sperm maturation. Additionally, sperm counts and sperm motility (percentage of sperm with progressive mobility and sperm with mobility) detected using CASA were significantly higher (*p* < 0.01) in the EXO group than that in the CON group (Figure 3E), indicating that hUMSC-Exos treatment improved fertility in aged mice.

Moreover, we detected the mRNA transcription levels of germ cell markers at different spermatogenetic stages, including genes of spermatogonia (*Dazl*, *Uchl1*, *Utf1*), spermatocytes (*Sycp3*, *Tex101*, *Pttg1*), round spermatocytes (*Acrv1*, *Tssk1*, *Spaca1*), and elongating spermatids (*Tnp2*, *Prm2*, *Best1*). We found that relevant mRNA expressions were upregulated after hUMSC-Exos treatment (Figure 3F), indicating the ability of hUMSC-Exos to restore spermatogenesis in aged testis.

### 3.4. hUMSC-Exos Treatment Promotes Testosterone Secretion in Aged Mice

It has been established that aged mice have lower serum testosterone levels than young mice [47], which prompted us to investigate the effect of hUMSC-Exos treatment on age-related testosterone deficiency. Compared with those in the CON group, serum testosterone levels of the EXO group were slightly higher (*p* < 0.05) on the 20th day (3.258 ± 0.177 ng/mL vs. 2.525 ± 0.120 ng/mL), but significantly higher (*p* < 0.01) on the 40th day (4.575 ± 0.215 ng/mL vs. 2.608 ± 0.195 ng/mL) (Figure 4A). It is currently believed that the decline in testosterone levels during aging is mainly due to the reduced activity of key enzymes in the steroidogenic pathway and a decreased number of Leydig cells (LCs). Accordingly, we examined the expression of steroidogenic enzymes. Notably, the expression level of steroid 17-alpha-hydroxylase/17,20 lyase (CYP17A1) and the number of hydroxy-delta-5-steroid dehydrogenase, 3 beta-, and steroid delta-isomerase 1 (HSD3B1)-positive cells in the EXO group were both higher (*p* < 0.001) than that in the CON group (Figure 4B,C), indicating that hUMSC-Exos treatment improved the function and number of LCs.

Furthermore, we also examined the weight of the levator ani/bulbocavernosus (LA/BC) muscles and the average muscle fiber size of the tibialis anterior muscle at the 40th day to elucidate the histological changes of androgen-responsive tissues. Compared with the CON group, hUMSC-Exos treatment significantly increased (*p* < 0.001) the weight of the LA/BC muscles (0.105 ± 0.002 g vs. 0.081 ± 0.003 g) and the average muscle fiber size of the tibialis anterior muscle in the EXO group (1833.33 ± 47.27 μm^2^ vs. 1183.10 ± 41.14 μm^2^) (Figure 4D,E). Additionally, the forelimb grip strength of aged mice in the EXO group was significantly higher (*p* < 0.01) than that in the CON group at the 40th day (Figure 4F). Finally, we found the increased mRNA expressions of LCs marker genes (*Star*, *Hsd3b1*, *Cyp11a1*, *Hsd17b3*, *Cyp17a1*, *Nr5a1*, *Insl3*), indicating that hUMSC-Exos treatment could promote testosterone synthesis at the molecular level (Figure 4G).

Summarily, these data suggest that hUMSC-Exos treatment can alleviate testosterone deficiency in aged mice.

### 3.5. hUMSC-Exos Treatment Modulates Macrophage Polarization

MSC-derived exosomes can promote M1-to-M2 macrophage polarization and increase anti-inflammatory cytokine levels [48]. In this study, the FACS showed that hUMSC-Exos treatment significantly increased (*p* < 0.001) the ratio of M2 macrophages (CD206-positive cells) to M1 macrophages (CD86-positive cells) (Figure 5A). Additionally, qRT-PCR analysis showed a decrease (*p* < 0.001) in the mRNA expression of proinflammatory genes (*Il-1b*, *Il-6*, *Tnf-α*, *Ccl2*, *Ccl7*, *Nos2*) and an increase in the mRNA expression of proinflammatory genes (*Il-4*, *Il-10*, *Ccl17*) after hUMSC-Exos treatment (Figure 5B,C). These results indicate that hUMSC-Exos treatment can downregulate the inflammatory level in aged testis.

As inflammation and oxidative stress are closely related, we next investigated whether hUMSC-Exos treatment could alleviate oxidative damage in aged testis. Notably, ROS levels in the testicular tissues of the EXO group were lower than that in the CON group after hUMSC-Exos treatment (Figure 5D). Additionally, hUMSC-Exos treatment significantly decreased (*p* < 0.001) the expression level of the oxidative damage biomarker 8-hydroxy-2′-deoxyguanosine (8-OHdG), especially in the seminiferous tubules, compared with that in the CON group (Figure 5E). These data suggest that hUMSC-Exos treatment can reduce oxidative stress by suppressing ROS production in aged testis.

### 3.6. Bioinformatic Analysis of hUMSC-Exos miRNA Expression

To further explore the potential mechanisms underlying the rescue of testicular aging by hUMSC-Exos, hUMSC-Exos miRNA expression microarray GSE69909 downloaded from the GEO database was utilized for bioinformatics analysis. We identified a total of 231 differentially expressed genes (DEGs) (logFC > 2 and *p*-value < 0.05), with 155 upregulated and 76 downregulated DEGs using edgeR package analysis (Figure 6A). Subsequently, we utilized a volcano plot and heatmap to highlight the top 10 significantly upregulated miRNAs, including hsa-miR-125b-5p, hsa-miR-100-5p, hsa-miR-199a-3p, hsa-miR-1260b, hsa-miR-145-5p, hsa-miR-31-5p, hsa-miR-146a-5p, hsa-miR-29a-3p, hsa-miR-376c-3p, hsa-miR-130a-3p (Figure 6A,B). Furthermore, we performed GO analysis and KEGG analysis for the 10 upregulated miRNAs to elucidate their potential molecular functions and associated pathways. The GO analysis indicated that these miRNAs are primarily associated with functions such as T cell differentiation, response to steroid hormone, response to oxidative stress, response to hydrogen peroxide, regulation of lipid kinase activity, regulation of carbohydrate catabolic process, regulation of autophagy, regeneration, natural killer cell differentiation, intrinsic apoptotic signaling pathway, gonad development, glycerophospholipid metabolic process, immunocytokines, cellular senescence, and cell growth (Figure 6C). Similarly, the KEGG analysis suggested that these miRNAs are primarily involved in pathways such as the TNF signaling pathway, T cell receptor signaling pathway, mTOR signaling pathway, Ras signaling pathway, PI3K-Akt signaling pathway, p53 signaling pathway, Notch signaling pathway, MAPK signaling pathway, lipid and atherosclerosis, insulin signaling pathway, growth hormone synthesis and secretion, GnRH signaling pathway, cellular senescence, autophagy, and apoptosis (Figure 6D). These findings suggest that hUMSC-Exos may exert the effect on delaying testicular aging through their miRNAs.

## 4. Discussion

In this study, we isolated exosomes from the supernatant of hUMSCs using a series of ultracentrifugation procedures, and the TEM results showed the hallmark exosome characteristic of a cup-like morphology. The particle size of the isolated hUMSC-Exos was approximately 139.6 nm, which is between 30 and 200 nm [49]. Exosomal markers CD9 and TSG101 were also expressed in the isolated hUMSC-Exos. These were in line with the standard characteristics of exosomes in previous reports [38]. Subsequently, we found that CM-Dil-labeled hUMSC-Exos were mainly localized in the testicular stroma after local testicular injection, with a small distribution observed at the periphery of the seminiferous tubules. In our study, we reported the distribution of hUMSC-Exos in the testis after injection, which has not been reported in other articles previously. This suggests that hUMSC-Exos can penetrate the blood–testis barrier and act on the cells both in testicular interstitium and spermatogenic tubules, potentially influencing the secretion of testosterone and altering spermatogenic function. Summarily, our results showed the typical exosomal characteristics of the extracted hUMSC-Exos and their dynamic distribution after testicular injection.

Referring to the previous reports, one of the characteristics of aging testis the reduction of volume and weight [50,51]. In our study, we revealed that hUMSC-Exos treatment enhanced both the testicular weight index and its morphological features, implying a potential regeneration of testicular tissue and rejuvenation of senescent cells. Within aging tissues, cells entering senescence exhibit pronounced cell cycle arrest and simultaneously secrete a diverse array of senescence-associated secretory phenotypes (SASP) [52,53]. These SASP factors have the capacity to induce senescence in adjacent cells, further compromising tissue functionality [52,53]. Our study found that hUMSC-Exos treatment suppressed the activity of proinflammatory SASP in senescent testicular tissues, specifically targeting IL-1b, IL-6, TNF-α, CCL2, and CCL7. Additionally, the anti-inflammatory effects of IL-4, CCL17, and IL-10 were enhanced in these tissues. In support of the hypothesis that hUMSC-Exos treatment can rejuvenate aged testis, we further observed an increase in LAMINB1 expression and a concomitant reduction in SA-β-Gal activity following hUMSC-Exos treatment. LAMINB1 is a major component of the nuclear lamina, and loss of LAMINB1 is a hallmark of cellular senescence [54]. During aging, SA-β-gal activity increases and senescent cells with high SA-β-gal activity appeared blue after staining [55]. The changes in LAMINB1 and SA-β-gal activity pointed toward a rejuvenation process within testicular cells. Further insights from our study, including the observed decrease in cell apoptosis post-treatment, underscore the potential of hUMSC-Exos in promoting cell survival and countering cell senescence. This notion was further bolstered by the elevated expression of PCNA following treatment. PCNA can serve as a cofactor of DNA polymerase to promote DNA localization, further participate in DNA synthesis and repair, and promote DNA replication. Since DNA replication indicates cell proliferation, the PCNA expression level reflects the proliferation status of cells [56,57]. In brief, these changes suggested that hUMSC-Exos treatment effectively relieved testicular senescence-associated phenotype and promoted regeneration of testicular tissue.

As hUMSC-Exos were mainly distributed in the periphery of the seminiferous tubules after injection, we hypothesized that they mainly acted on SSCs after crossing the blood–testis barrier. SSCs are crucial for spermatogenesis and are localized within the basement membrane of the seminiferous tubules in adult testes [58]. They possess an inherent capacity for self-renewal, ensuring a consistent population of stem cells participating in spermatogenesis [58]. Our research has elucidated that hUMSC-Exo intervention promotes the proliferation of SSCs, as evidenced by increased expression levels of SSCs markers, namely STRA8. As an ATP-dependent RNA helicase, DDX4 is considered a germ cell marker and is involved in germ cell development, proliferation, and differentiation [59], and the upregulation of DDX4 indicated that hUMSC-Exos intervention promoted the differentiation of SSCs into mature germ cells. In addition, PNA can selectively bind to the acrosomes of sperm, which is a marker of haploid spermatids for the evaluation of mature sperm number and integrity [60]. Therefore, increased PNA expression and the results of semen analysis further support the proposition that hUMSC-Exos treatment has the potential to restore fertility in aged mice. In summary, hUMSC-Exos treatment has the efficacy of promoting spermatogenesis.

After injection, hUMSC-Exos also abundantly distributed in the testicular interstitium. As the major cells in the interstitial region of testis, LCs synthesized and secreted more than 95% of testosterone in the male serum [4,7]. The prevalent consensus links the testosterone decline observed in aging to a decrease in essential enzymes within the steroidogenic pathway and a reduction in LCs numbers [4,7]. Remarkably, our findings indicate that hUMSC-Exos treatment upregulated the expression of steroid synthase and augmented the LCs population, culminating in a pronounced increase in serum testosterone in aged mice. These data suggest that hUMSC-Exos might not only enhance the functionality of LCs but also activate LCs progenitor cells, guiding their differentiation into mature LCs. In addition, deficits in testosterone in the aging population correlate with reduced energy, attenuated muscle strength, and overall physiological deterioration [4,7]. In this context, our study further revealed that hUMSC-Exos treatment ameliorated the architecture of androgen-responsive tissues and amplified muscle function, suggesting that hUMSC-Exos might be instrumental in mitigating systemic manifestations of testosterone insufficiency. In brief, hUMSC-Exos treatment could improve the function of steroid synthesis of aged testis and alleviate the symptoms of testosterone deficiency.

There is an immunosuppressive environment in the testis that is critical in ensuring optimal steroidogenesis, spermatogenesis, and other testicular homeostatic functions [61]. Within the testes, macrophages represent the primary immune cell type, and their dysfunction is intimately associated with age-induced inflammation [62]. Notably, aged testis demonstrates a heightened prevalence of proinflammatory M1 macrophages. These macrophages release inflammatory cytokines that disrupt tissue equilibrium and hinder testicular functionality [63]. Our findings revealed that intervention with hUMSC-Exos catalyzed the transition of M1 macrophages toward the M2 phenotype in aged testis. This transition was underscored by a decrease in proinflammatory genes (*Il-1b*, *Il-6*, *Tnf-α*, *Ccl2*, *Ccl7*, *Nos2*) and an increase in anti-inflammatory genes (*Il-4*, *Il-10*, *Ccl17*). Equally significant is the fact that certain mediators like TNFα, CCL7, and IL-1b can negatively affect the proliferative and differentiative roles of SSCs, as well as testosterone synthesis in LCs. Conversely, cytokines such as IL-4 and IL-10 are vital in establishing the immune stability needed for regular spermatogenesis [64]. In addition, considering the testis’ significant vulnerability to ROS-induced damage and the compound effect of ROS and inflammation on testicular function [65], we further detected the change in ROS post-hUMSC-Exos treatment. The marked reduction in ROS level and expression of ROS damage marker 8-OHdG treatment underscored a decline in oxidative stress. In summary, these revelations suggest that hUMSC-Exos can balance the chaotic state of inflammatory and oxidative stress in aged testis.

Mechanistically, MSC-Exos encapsulate a diverse range of active cellular components, including lipids, proteins, mRNAs, tRNAs, lncRNAs, miRNAs, and circRNAs [18]. To our knowledge, miRNAs are crucial players during normal development, homeostasis, and disease, which participate in almost every biological process. As miRNAs are abundant in hUMSC-Exos, we hypothesized that hUMSC-Exos promoted testosterone synthesis mainly through their miRNAs. Therefore, we performed additional bioinformatics analysis for hUMSC-Exos miRNAs. Notably, has-miR-125b-5p, has-miR-100-5p, has-miR-145-5p, and has-miR-146a-5p, enriched in hUMSC-Exos miRNAs, have been reported to be closely related to testosterone synthesis [66,67,68,69]. The has-miR-145-5p, has-miR-31-5p, and has-miR-199a-3p, enriched in hUMSC-Exos miRNAs, have been reported to participate in regulation of spermatogenesis [70,71,72]. Referencing previous reports, these miRNA-associated molecular functions and signaling pathways in the GO and KEGG analysis are also closely related to testosterone synthesis, inflammatory response, oxidative stress, cell senescence, etc. [73,74,75,76,77], suggesting that the rescue of testicular aging using hUMSC-Exos treatment may be mediated by hUMSC-Exos miRNAs.

There are several limitations to this study that warrant further investigation. First, although this study implies that miRNAs is one of the essential components in hUMSC-Exos responsible for restoring testis function and rescuing testicular aging, we cannot exclude the potential role of other secretory components such as proteins, lipids, or other RNAs that could contribute to testicular rejuvenation. Referencing previous reports, enzymes that are pivotal to the ATP-producing phase of glycolysis, namely, phosphoglycerate kinase (PGK), phosphoglucomutase (PGM), enolase (ENO), and pyruvate kinase m2 subtype (PKm2), are present within MSC-Exos. These enzymes may play a role in augmenting energy metabolism to rejuvenate aging cells [78,79]. In light of this intricacy, the mechanism behind hUMSC-Exos therapy needs thorough examination. Second, we have not yet intensively investigated the specific molecular mechanism and signaling pathways by which miRNAs mediate the efficacy of hUMSC-Exos therapy. Third, species difference must be considered for clinical application. The safety and efficacy assessments for hUMSC-Exos treatment in primates and cultured human testicular tissue are imperative in the future. Last, another challenge in clinical application of hUMSC-Exos therapy for testicular aging is the standardization in the preparation process. To ensure consistent therapeutic effects, standardized methods for harvesting, purifying, and storing hUMSC-Exos are crucial and need further investigation.

## 5. Conclusions

In conclusion, our results reveal that hUMSC-Exos administration exhibited potent anti-aging efficacy in aged testis, notably augmenting spermatogenesis and testosterone synthesis, and effectively suppressing inflammation and attenuating oxidative stress. These beneficial effects may be mediated by hUMSC-Exos miRNAs. Taken together, these data highlight the potential of hUMSC-Exos in rescuing testicular aging, suggesting their important utility as a therapeutic strategy in this domain.

## Figures and Tables

**Figure 1 biomedicines-12-00098-f001:**
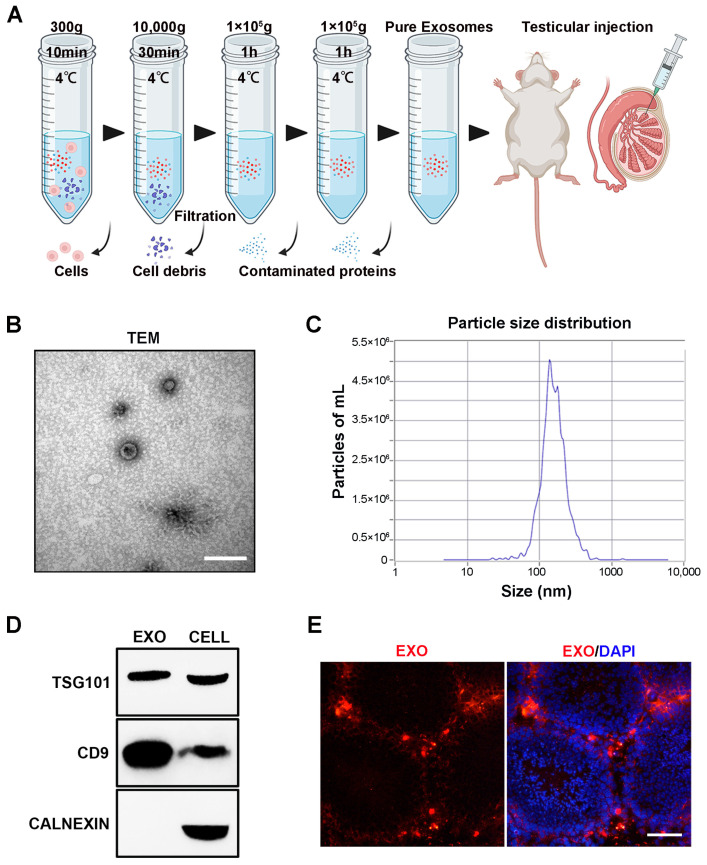
Isolation, identification, characterization, and tracking of hUMSC-Exos. (**A**) Flowchart of the method used for hUMSC-Exos purification via differential ultracentrifugation. (**B**) Representative TEM image showing hUMSC-Exos morphology. Scale bar = 100 nm. (**C**) Particle size distribution of isolated hUMSC-Exos, as determined using NTA. (**D**) Analysis of the expression of exosome markers (TSG101 and CD9) in hUMSC-Exos using Western blotting. The picture shows a cropped image of the blot, and the uncropped full-length gels and blots are shown in Appendix A. (**E**) Fluorescence microscopy image of testicular sections from aged mice showing the presence of transplanted CM-Dil-labeled (red) hUMSC-Exos. DAPI (blue) was used to indicate the nucleus. Scale bar = 50 μm.

**Figure 2 biomedicines-12-00098-f002:**
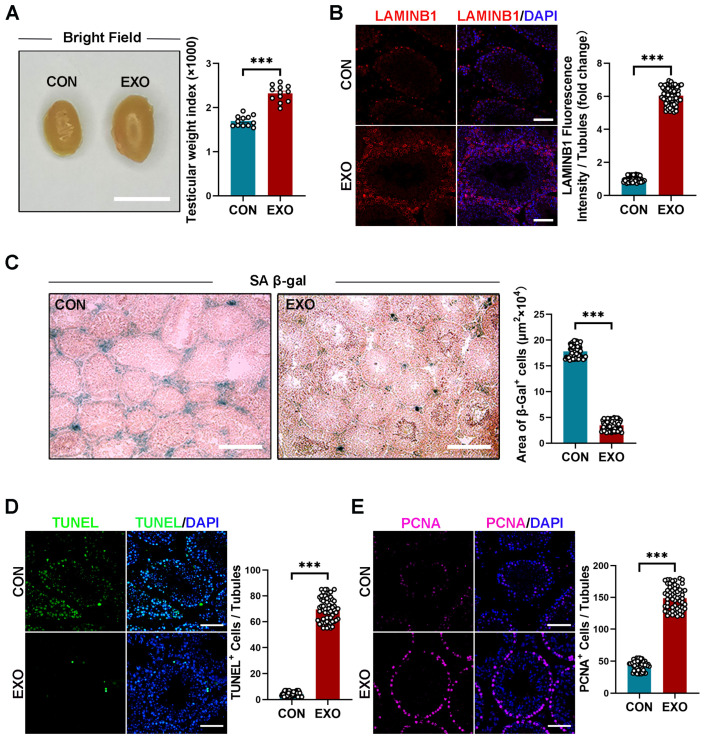
hUMSC-Exos treatment ameliorated cellular senescence in aged testis. (**A**) Left: representative light images of testes from the CON and EXO groups. Right: quantification of the testicular weight indexes (testicular weight/body weight × 1000) of bilateral testes of each mouse. Points represent 12 independent weight indexes per group (CON: *n* = 6 mice; EXO: *n* = 6 mice). Scale bar = 1 cm. (**B**) Left: representative images of LAMINB1 of the CON and EXO groups. Right: quantification of the expression level of LAMINB1. The ratio between fluorescence intensity of LAMINB1+ cells (red) and number of seminiferous tubules in one vision was calculated. Points represent the ratios of 60 independent visions per group (CON: *n* = 6 mice; EXO: *n* = 6 mice). DAPI (blue) was used to indicate the nucleus. Scale bar = 50 μm. (**C**) Left: representative images of SA-β-gal staining of the CON and EXO groups. Right: quantification of the area of SA-β-gal+ cells per vision. Points represent the areas of 60 independent visions per group (CON: *n* = 6 mice; EXO: *n* = 6 mice). Scale bar = 100 μm. (**D**) Left: representative images of TUNEL staining of the CON and EXO groups. Right: quantification of the number of TUNEL+ cells per vision. The ratio between number of TUNEL+ cells (green) and number of seminiferous tubules in one vision was calculated. Points represent the ratios of 60 independent visions per group (CON: *n* = 6 mice; EXO: *n* = 6 mice). DAPI (blue) was used to indicate the nucleus. Scale bar = 50 μm. (**E**) Left: representative images of PCNA in the CON and EXO groups. Right: quantification of the number of PCNA+ cells per vision. The ratio between number of PCNA+ cells (purple) and number of seminiferous tubules in one vision was calculated. Points represent the ratios of 60 independent visions per group (CON: *n* = 6 mice; EXO: *n* = 6 mice). DAPI (blue) was used to indicate the nucleus. Scale bar = 50 μm. *** *p* < 0.001.

**Figure 3 biomedicines-12-00098-f003:**
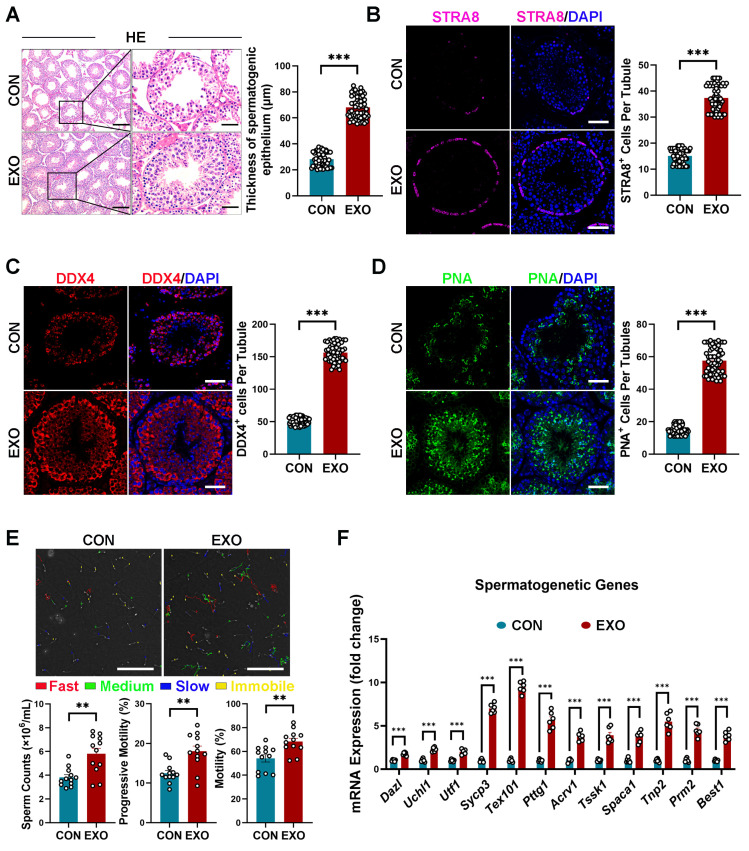
hUMSC-Exos treatment improved spermatogenesis. (**A**) Left: representative light micrographs of testis sections stained with H&E from the CON and EXO groups. Right: quantification of thicknesses of the spermatogenic epithelium per tubule. Points represent the quantitative results of 60 independent tubules per group (CON: *n* = 6 mice; EXO: *n* = 6 mice). Scale bars: left—100 μm; right—50 μm. (**B**) Left: representative images of STRA8 (purple) of the CON and EXO groups. Right: quantification of the number of STRA8+ cells per tubule. Points represent the quantitative results of 60 independent tubules per group (CON: *n* = 6 mice; EXO: *n* = 6 mice). DAPI (blue) was used to indicate the nucleus. Scale bar = 50 μm. (**C**) Left: representative images of DDX4 (red) of the CON and EXO groups. Right: quantification of the number of DDX4+ cells per tubule. Points represent the quantitative results of 60 independent tubules per group (CON: *n* = 6 mice; EXO: *n* = 6 mice). DAPI (blue) was used to indicate the nucleus. Scale bar = 50 μm. (**D**) Left: representative images of PNA (green) of the CON and EXO groups. Right: quantification of the number of PNA+ cells per tubule. Points represent the quantitative results of 60 independent tubules per group (CON: *n* = 6 mice; EXO: *n* = 6 mice). DAPI (blue) was used to indicate the nucleus. Scale bar = 50 μm. (**E**) Top: representative images of CASA analysis of the CON and EXO groups. Bottom: quantification of the sperm parameters (sperm counts, percentage of sperm with progressive mobility, and percentage of sperm with mobility). Points represent the obtained values of 12 independent CASA analyses per group (CON: *n* = 6 mice; EXO: *n* = 6 mice). Scale bar = 100 μm. (**F**) Quantification of spermatogenic gene expression in the CON and EXO groups (CON: *n* = 6 mice; EXO: *n* = 6 mice). ** *p* < 0.01, and *** *p* < 0.001.

**Figure 4 biomedicines-12-00098-f004:**
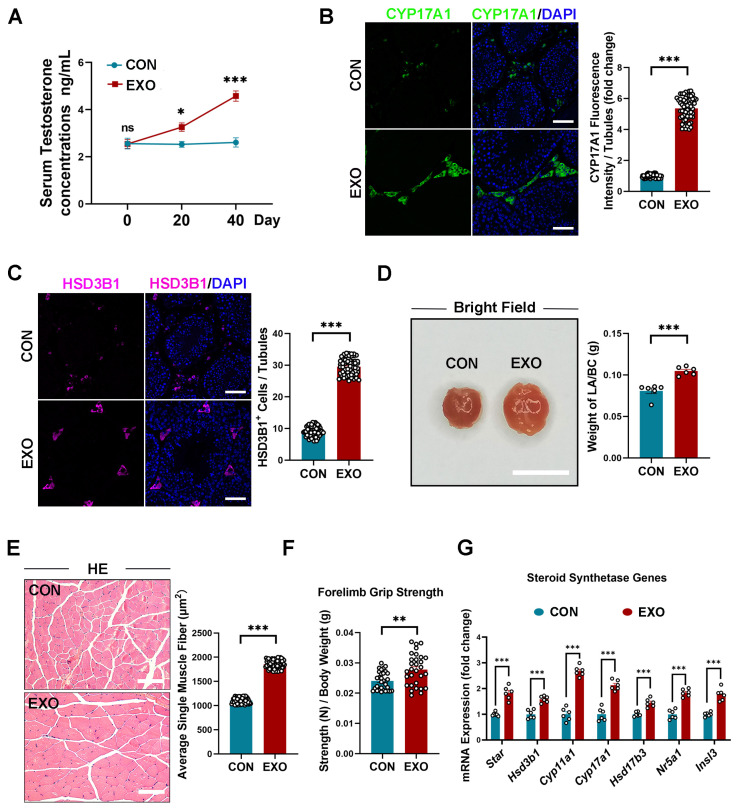
hUMSC-Exos treatment promoted testosterone synthesis and secretion in naturally aged mice. (**A**) Quantification of serum testosterone levels in CON and EXO groups. Bars represent the mean with sem of six independent testosterone values per group (CON: *n* = 6 mice; EXO: *n* = 6 mice). (**B**) Left: representative images of CYP17A1 of the CON and EXO groups. Right: quantification of the expression level of CYP17A1. The ratio between fluorescence intensity of CYP17A1+ cells (green) and number of seminiferous tubules in one vision was calculated. Points represent the ratios of 60 independent visions per group (CON: *n* = 6 mice; EXO: *n* = 6 mice). DAPI (blue) was used to indicate the nucleus. Scale bar = 50 μm. (**C**) Left: representative images of HSD3B1 of the CON and EXO groups. Right: quantification of the number of HSD3B1+ cells. The ratio between number of HSD3B1+ cells (purple) and number of seminiferous tubules in one vision was calculated. Points represent the ratios of 60 independent visions per group (CON: *n* = 6 mice; EXO: *n* = 6 mice). DAPI (blue) was used to indicate the nucleus. Scale bar = 50 μm. (**D**) Left: representative light images of LA/BC muscles of the CON and EXO groups. Right: quantification of the weight of the LA/BC muscles. Points represent the values of six independent weights per group (CON: *n* = 6 mice; EXO: *n* = 6 mice). Scale bar = 1 cm. (**E**) Representative light micrographs of tibialis anterior muscle stained with H&E from the CON and EXO groups. Right: quantification of the average single muscle fiber size (total muscle fiber area/total muscle fiber number) of each vision. Points represent the average single muscle fiber sizes of 60 independent visions per group (CON: *n* = 6 mice; EXO: *n* = 6 mice). Scale bar = 50 μm. (**F**) Quantification of the forelimb grip strength of the CON and EXO groups. Points represent 30 strength values from five repeated experiments per group (CON: *n* = 6 mice; EXO: *n* = 6 mice). (**G**) Quantification of steroid synthesis gene expression in the CON and EXO groups (CON: *n* = 6 mice; EXO: *n* = 6 mice). ns = not significant. * *p* < 0.05, ** *p* < 0.01, and *** *p* < 0.001.

**Figure 5 biomedicines-12-00098-f005:**
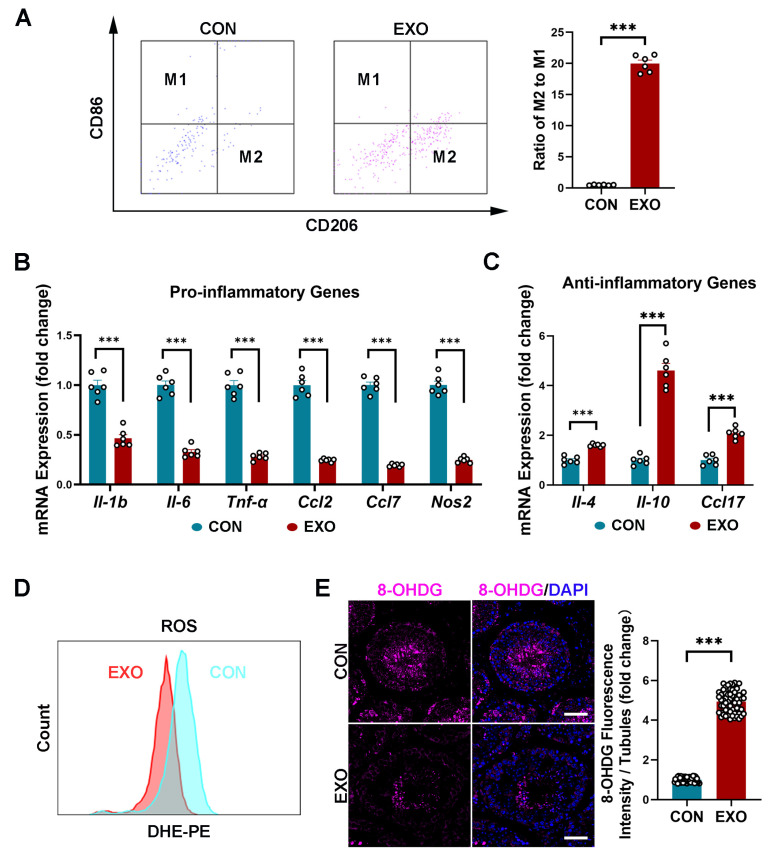
hUMSC-Exos treatment modulated macrophage polarization and reduced oxidative stress. (**A**) Left: representative images of FACS analysis of CON (blue) and EXO (purple) groups. Right: quantification of the ratio of M2 to M1 macrophages. Bars represent the mean with sem of six independent ratios per group (CON: *n* = 6 mice; EXO: *n* = 6 mice). (**B**) Quantification of proinflammatory gene expression in the CON and EXO groups (CON: *n* = 6 mice; EXO: *n* = 6 mice). (**C**) Quantification of anti-inflammatory gene expression in the CON and EXO groups (CON: *n* = 6 mice; EXO: *n* = 6 mice). (**D**) Quantification of ROS production in CON and EXO groups, as measured using flow cytometry. CON: *n* = 6 mice; EXO: *n* = 6 mice. (**E**) Left: representative images of 8-OHdG of the CON and EXO groups. Right: quantification of the number of 8-OHdG+ cells per vision. The ratio between number of 8-OHdG+ cells (purple) and number of seminiferous tubules in one vision was calculated. Points represent the ratios of 60 independent visions per group (CON: *n* = 6 mice; EXO: *n* = 6 mice). DAPI (blue) was used to indicate the nucleus. Scale bar = 50 μm. *** *p* < 0.001.

**Figure 6 biomedicines-12-00098-f006:**
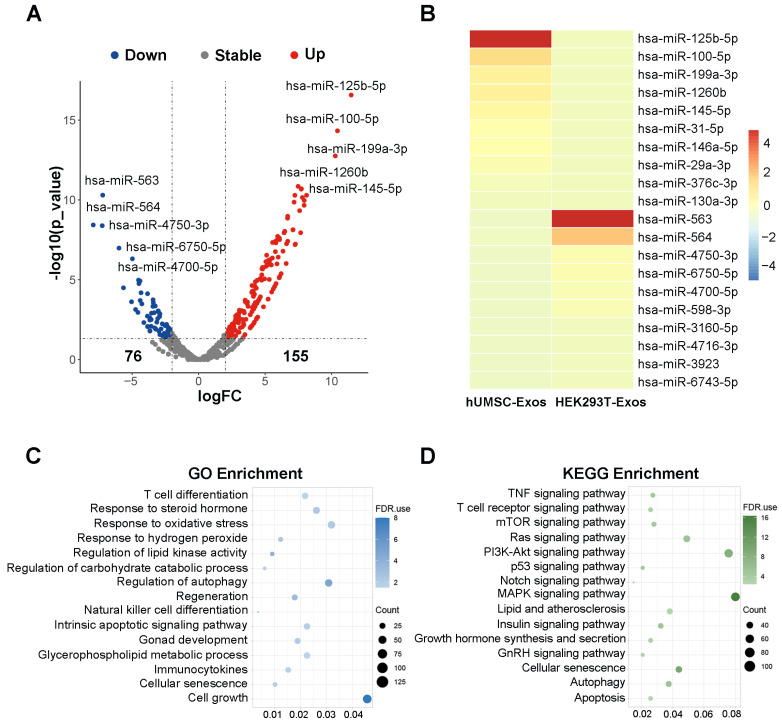
(**A**) Volcano plot identifying a total of 231 DEGs, with 155 upregulated and 76 downregulated DEGs. (**B**) Heatmap illustrating the top 10 upregulated DEGs and the top 10 downregulated DEGs. (**C**) GO enrichment analysis of the top 10 upregulated DEGs. (**D**) KEGG enrichment analysis of the top 10 upregulated DEGs.

## Data Availability

The datasets used and/or analyzed during the current study are available from the corresponding author upon reasonable request.

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
