# Peer review of "Human Umbilical Cord Mesenchymal Stem Cell-Derived Exosomes Rescue Testicular Aging"

_biomedicines, 2024, doi:10.3390/biomedicines12010098_

Round 1
Reviewer 1 Report
Comments and Suggestions for Authors
The manuscript describes the beneficial effect of the intratesticular injection of exosomes from MSCs on testicular aging. In general, the manuscript requires further experiments to support the conclusions (more quantitative methods). In addition, the methods require further description and clarification. In particular:
1. Section 2.2. The method of intratesticular injection requires further explanation. Which type of injection system was used? Reference 12 (Staff et al.,) does not describe any form of intratesticular injection.
2. Section 2.5.2. Please, provide the camera level and threshold used for nanoparticle tracking analysis.
3. Section 2.5.3. Please, confirm the density of the polyacrylamide gel. 4% for small proteins such as CD9 and TSG101 seems too low.
4. Section 2.4. The authors provided Reference 16 (Yang et al.) as reference for exosome isolation. However, this reference is for microvesicles. Important details such as the number of cells need to be provided in the materials and methods.
5. Lines 231-232. It is not clear, how a nanodrop device can provide information on the RNA integrity
6. It is not clear how the signal from fluorescence microscopy images was quantified. Which software did the authors use? How did the authors distinguish between positive and negative cells? The authors should perform another quantitative method to confirm the findings obtained by fluorescence microscopy.
7. Lines 348-349. The reference 28 is not correct (does not describe a study in mice but, instead, in humans).
8. In order to support the conclusions, results from Fig. 4 (CYP17A1 and HSD3B1 expression) need to be confirmed by another method (e.g., western blot or qRT-PCR).
Reviewer 2 Report
Comments and Suggestions for Authors
In the manuscript titled "Human Umbilical Cord Mesenchymal Stem Cell-Derived Exosomes Rescue Testicular Aging," the authors aim to explore the therapeutic potential of exosomes derived from human umbilical cord mesenchymal stem cells in mitigating the effects of testicular aging. While the topic is intriguing, the paper falls short of its promise due to several significant drawbacks. These flaws hinder the scientific integrity of the content. I have specific comments that could contribute to refining the manuscript.
Specific comments:
1. The introduction lacks a clear and comprehensive rationale for the chosen experimental approach. The identification of the gap in the existing literature and the ways in which this study aims to address and fill that gap are not sufficiently elucidated. A more thorough context and rationale are needed to guide the reader seamlessly into the significance of the research.
2. Regarding Figure 1, it seems that the Western blot bands, especially those related to CD9, may have undergone manipulation. This observation could be attributed to the quality of the provided version. I recommend a meticulous examination to ensure the accuracy and integrity of the figures.
3. The examination of spermatogenesis in the study is not exhaustive enough to draw definitive conclusions. I would recommend the authors to broaden their investigation into the integrity of spermatogenesis for a more comprehensive and conclusive analysis.
4. The authors assert that HUMSC-Exo treatment enhances testosterone secretion in the aged testis. However, the mechanism through which this is achieved remains unclear. A more detailed explanation of the processes and pathways involved in this observed effect would greatly enhance the clarity and understanding of the study's findings.
5. Inflammation is a potential outcome of the treatment and should be thoroughly examined to understand its implications.
6. The discussion section is currently lacking in strength and cohesiveness. The limitations are neither adequately discussed nor clearly presented. Furthermore, a conclusive takeaway message is absent. Strengthening the discussion, addressing limitations, and providing a clear takeaway message would significantly enhance the overall impact and coherence of the paper.
Comments on the Quality of English Languageminor changes required
Reviewer 3 Report
Comments and Suggestions for Authors
This is a very interesting study showing that injections of exosomes released from human MSCs into the testes of aged mice increase testosterone production and restore spermatogenesis. The interpretation of the results is limited because only one strain of mice was used and the study was conducted with the minimum number of animals required, but this is unavoidable in view of animal welfare. There is no mention of the limitations of this study, but this is not a problem as there are no excessive claims in the discussion and conclusions. I am satisfied that the analytical methodology covers all the necessary points.
There is almost no problem with this as it is for publication, but I would like you to confirm the following two points.
1)This was probably an accident during PDF conversion, but there is a probably unnecessary character "Table 1.A)" on line 258.
2)At the end of the results, the authors state, “Overall, these data suggest that hUMSC-Exo treatment can reduce oxidative stress by modulating macrophage polarization in aged testes.” However, as the experimental design does not appear to be a time series of samples, I do not believe it is possible to determine from these results whether macrophage polarization occurred first and oxidative stress was subsequently reduced. The authors should indicate if they have any evidence on this point.
Round 2
Reviewer 1 Report
Comments and Suggestions for Authors
The authors have addressed all my concerns and delivered a significantly improved version.
Reviewer 2 Report
Comments and Suggestions for Authors
The paper is acceptable